# A “Mammalian-like” Pycnodont Fish: Independent Acquisition of Thecodont Implantation, True Vertical Replacement, and Carnassial Dentitions in Carnivorous Mammals and a Peculiar Group of Pycnodont Fish

**DOI:** 10.3390/life12020250

**Published:** 2022-02-08

**Authors:** Kumiko Matsui, Yuri Kimura

**Affiliations:** 1Department of Paleobiology, National Museum of Natural History, Smithsonian Institution, Washington, DC 20001, USA; MatsuiK@si.edu; 2The Kyushu University Museum, Kyushu University, Fukuoka 812-8581, Fukuoka, Japan; 3Department of Geology and Paleontology, National Museum of Nature and Science, Tsukuba 305-0005, Ibaraki, Japan; 4Institut Català de Paleontologia Miquel Crusafont, ICTA-ICP, Campus de la Universitat Autònoma de Barcelona, Edifici Z. Carrer de les Columnes, s/n, E-08193 Cerdanyola del Vallès, Barcelona, Spain

**Keywords:** tooth replacement, pycnodont, serrasalmimid, carnassial teeth, K-Pg mass extinction, carnivorous mammals

## Abstract

Vertebrates developed tooth replacement over 400 million years ago. Then, 200 million years later, the combination of vertical tooth replacement with the thecodont implantation (teeth in bone sockets) appeared a key morphological innovation in mammalian evolution. However, we discovered that an extinct fish taxon, *Serrasalmimus secans*, showed the same innovation in the lineage Serrasalmimidae, which survived the end Cretaceous mass extinction event. The carnassial teeth are known in both mammals and pycnodont fish, but these teeth do not share the same tissues or developmental processes. Therefore, this serrasalmimid pycnodont fish might have independently acquired mammal-like tooth replacement and implantation, indicating that the fish and mammals convergently evolved the carnassial dental morphologies at about the same time, approximately 60 My ago, in separate ecosystems.

## 1. Introduction

Vertebrates first acquired a tooth replacement mechanism during the latest Silurian, 424 million years ago [1], which played a key role in shaping the predator-prey relationships over time. Since that time, vertebrates have evolved numerous patterns of tooth replacement and attachment e.g., [2]. In general, in Chondrichthyes, Osteichthyes, Amphibia, and Reptilia, teeth are replaced multiple times during their lifetime, whereas in mammals, teeth are shed only once or less e.g., [3]. The teeth are implanted on the lingual side (i.e., pleurodont) or on the margin (i.e., acrodont) in teleosts, amphibians, and squamates [4], whereas mammals and archosaur reptiles have their tooth roots enclosed in the alveoli (i.e., ankylothecodonty) [2]. 

The pycnodont fishes, Pycnodontiformes, are an extinct clade of Actinopterygii, which lived from the Late Triassic to the late Eocene [5]. Within this clade, the family Serrasalmimidae is characterized by the reduced tooth rows in their ankylothecodont-like dentition (i.e., absence of true socket, but root-like structure firmly fused to the bone as in other pycnodonts). The geologically youngest taxon *Serrasalmimus secans*, Vullo et al. [6], known from the Paleocene in Morocco, possessed a cutting dentition and retained only a paired set of single tooth rows on both sides of the tooth-bearing bones (vomer for upper and prearticular for lower dentition) to enhance the shear force with labio-lingually compressed bicuspid teeth [6]. The compressed mammary-form teeth of *S. secans* show vertical wear facets by shearing on the labial side of the vomerine dentition and the lingual side of the prearticular dentition. This pattern resembles the carnassial teeth of mammals (though, in a reverse contact). In the present study, we characterized the combination of tooth replacement pattern and implantation in *S. secans* based on 3D reconstructions of tooth structures of functional teeth in *S. secans*. We hypothesize that serrasalmimids independently acquired a vertical replacement in true thecodont implantation, which is a characteristic tooth replacement pattern of mammals. 

## 2. Methods

### Computed Tomography

We scanned a specimen of *S. secans* (NMNS-PV20561) using X-ray micro-CT scanning with a beam energy of 100 kV and a flux of 100 μA at a resolution of 0.011 mm (SkyScan 1275 Micro-CT (Bruker, Billerica, MA, USA). The procedure was conducted at the Primate Research Institute of Kyoto University, Japan. The resulting scanned data were imported into Avizo Lite 9.3 and Amira 2020.2 (Thermo Fisher Scientific Inc., Waltham, MA, USA) for digital segmentation, rendering, and reconstruction of the internal structures.

## 3. Results and Discussion

Our analysis focused on a specimen of *S. secans* (Figure 1A, Appendix A), which was collected from the Phosphorite Bed II (Thanetian age, Paleocene) in the Ouled Abdoun Basin, Morocco. A three-dimensional reconstruction of the internal structures of NMNS-PV20561 showed the thecodont-like implantation in ankylosis attachment to the vomer bone (Figure 1C), as previously reported [6]. Furthermore, we visualized the internal structure of the vomer with dental implantations and the tooth replacement system (Figure 1B,C). A pulp cavity contained a tooth germ in the center of each functional tooth, in contrast to common fishes, in which their tooth germs are formed on the lingual side of the functional tooth [7]. NMNS-PV20561 did not show any traces of bone resorption (i.e., the replacement pore; see [8] for terminology) on the buccal and lingual sides of the bone, unlike other fishes that possess several tooth rows (e.g., wolf fishes and blue fishes [9]). In archosaurians (e.g., crocodiles and dinosaurs), the replacement teeth are formed on the lingual side of the functional teeth, and then they dissolve the wall of the functional tooth roots and move to a cavity below the functional teeth (see Figure 1 and [10]). In mammals, the replacement teeth and tooth germs are formed directly below functional teeth (Figure 1). From these observations, we suggest that the replacement teeth in *S. secans* would have developed between the tubular root structures of the functional teeth and erupted from their position directly below the functional teeth rather than their sides. This pattern closely resembled the vertical mode of tooth development in mammals. In addition, NMNS-PV20561 possibly possessed only one generation of tooth germs, and no buds or caps, under its functional teeth, which indicates that NMNS-PV20561 perhaps shed its teeth only once in its life.

In Mammalia, the carnassial dentition is a set of lingually bladed upper teeth and labially bladed lower teeth. They are specialized for shearing by a scissor-like action and leave an occlusal contact on the bladed surface of the tooth sides rather than towards the oral cavity, as observed for the non-carnassial premolars and molars. This specialized function has evolved by modifying the plesiomorphic tribosphenic molars and appears multiple times within different clades of mammals. Therefore, except for Eutriconodonta, it emphasizes the gradual adaptation to a carnivorous diet [11,12,13,14]. In the mammalian fossil record, the carnassial teeth appear in pretribosphenic Eutriconodonta and Metatheria (Thylacoleonidae, Dasyuridae, and Thylacinidae), and Eutriconodonta that do not have tribosphenic molars (Mesonychia, Oxyaenodonta, Hyaenodonta, Carnivoramorpha, and Sparassodonta) (Figure 2 and [14]). Previously, there were no single non-mammalian taxa with true carnassial teeth in combination with one-time replacement and the thecodont implantation. *S. secans* is considered as the first one with the carnivorous mammalian dental traits in non-mammalian clades.

In Mammalia, the earliest larger representative (about 10 kg body mass) with a dentition that emphasized the shearing function is the early Cretaceous *Repenomamus* with pretribosphenic triconodontan molars [15]. However, it is only after the end Cretaceous (i.e., K-Pg) mass extinction event that mammals acquired true carnassial molars with more precise shearing and a higher degree of morphological convergence for carnivory [14] (Appendix A). Cenozoic carnivorous mammals appeared worldwide approximately 60 Ma in the Paleocene (Figure 2; Appendix A). Notably, the earliest appearance of carnivorous mammals is contemporary to that of *S. secans* (60 Ma), six to seven million years (m.y.) after the K-Pg event (Figure 2). This period approximates the suggested duration of ecosystem restoration for metazoans after the K-Pg event (from 100 k.y. to 10 m.y) [16,17], which affected both the terrestrial and marine ecosystems [18,19,20,21]. The top predators that were wiped out in the extinction event included mosasaurs, plesiosaurs e.g., [22,23] and large predatory fishes (such as *Xiphactinus* [24] in the ocean and non-avian theropod dinosaurs on land). It has been proposed that these top trophic niches were not fully occupied until the late Paleogene [25], when the top predators in both the marine and terrestrial realms acquired functional carnassial teeth. 

*S. secans* is known for a diet different from that of other Pycnodontiformes. Almost all the species of Pycnodontiformes are known for durophagy e.g., [26,27], whereas *S. secans* is thought to be a flesh-eater that ate shell-less, soft-bodied animals such as cuttlefish, jellyfish, and small fishes [6]. Carnivorous mammals were absent in the Tethys Sea until the first appearance of early whales such as *Himalayacetus* in the early Eocene [28]. We suggest that the acquisition of carnassial teeth must be closely related to filling the ecospace vacated by the K-Pg mass extinction and that the predatorial niche was available to the serrasalmimid pycnodonts in the shallow ocean before the early whales evolved. 

Although *S. secans*, carnivorans, and some marsupials have carnassial teeth, the teeth of fish and mammals do not share the same dental tissues or developmental processes. For example, fish dentitions consist of enameloid and dentine, while mammalian teeth consist of enamel, dentine, and cement. On the other hand, enameloid contains collagen and is made by odontoblasts and dental epithelial cells between a well-defined dental papilla (mesenchyme) and enamel organ (epithelium) [29,30]. Alternatively, the enamel is an inorganic material produced by the epithelial cells interacting with mesenchymal cells in the underlying dental pulp [29,31]. Therefore, the carnassial teeth of *S. secans* and its mammals have different origins and developmental processes, representing a homoplastic convergence.

Currently, tooth implantation and replacement in fishes is an active research interest e.g., [32,33]. Some actinopterygians are known to have functional teeth with the thecodont attachment but no vertical replacement. For example, parrotfish (Teleostei: Scaridae) have thecodont implantation but possess multiple rows of teeth. The foremost teeth are used only once and then are lost and replaced by the successor teeth [7,34,35]. Our findings suggest that there might be more examples of homoplastic convergent evolution between phylogenetically distant vertebrate groups, such as the actinopterygians and mammals, which were not previously known. To the best of our knowledge, *S. secans* are the only actinopterygian identified so far with the vertical replacement mode of ankylothecodont teeth.

Here, we provide the first evidence of tooth replacement in pycnodont fishes. More importantly, we showed that the flesh-eating fish *Serrasalmimus secans* had a true vertical mode of tooth replacement with thecodont implantation, which was previously thought to be an exclusively mammalian characteristic. Both these lineages with carnassial teeth first appeared less than six to seven million years after the K-Pg event and likely reflected the concurrent filing of vacant top trophic niches in the marine and terrestrial realms. Although the tooth origins and developmental processes are very different between the two groups, they share some characteristics and functions.

## 4. Conclusions

We discovered that the extinct actinopterygian taxon *Serrasalmimus secans* shows the combination of mammalian-like vertical tooth replacement with the thecodont implantation. The serrasalmimid pycnodont fish specimen that we studied revealed independently acquired mammal-like tooth dentition features; this, therefore, indicates that fish and mammals evolved convergent carnassial dental morphologies despite being parts of separate ecosystems.

## Figures and Tables

**Figure 1 life-12-00250-f001:**
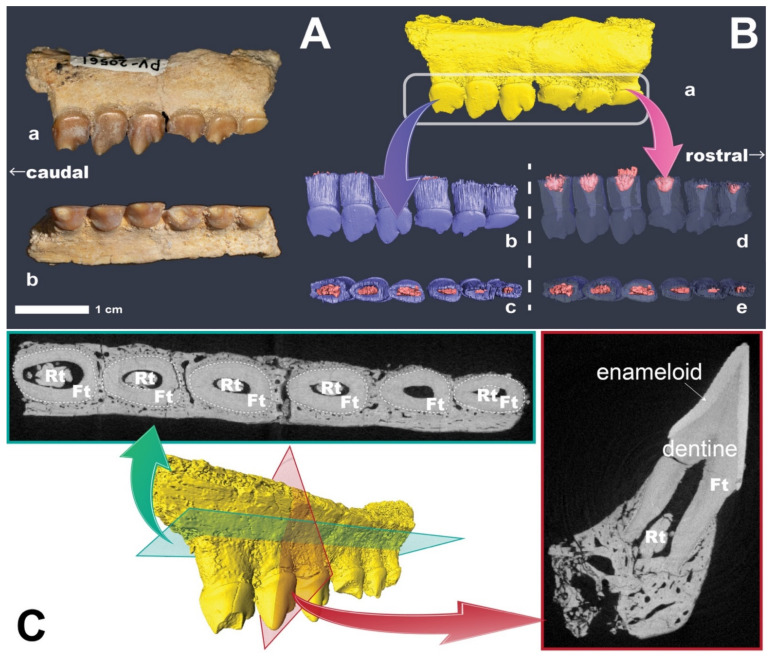
A right vomer of *Serrasalmimus secans* from the Phosphorite Bed II (Thanetian age, Paleocene) in the Ouled Abdoun Basin, Morocco. (**A**) A right vomer of S. secans, NSM-PV20561. (**a**) Lateral view. (**b**) Ventral view. (**B**) CT-based 3D models of NSM-PV20561. (**a**) Lateral view of NSM-PV20561 (yellow). (**b**) Lateral view of functional teeth (blue) with replacement teeth (pink). (**c**) Dorsal view of functional teeth with replacement teeth. (**d**) Lateral view of translucent functional teeth with replacement teeth. (**e**) Dorsal view of translucent functional teeth with replacement teeth. (**C**) Horizontal and longitudinal sections of NSM-PV20561. Rt: replacement teeth. Ft: functional teeth.

**Figure 2 life-12-00250-f002:**
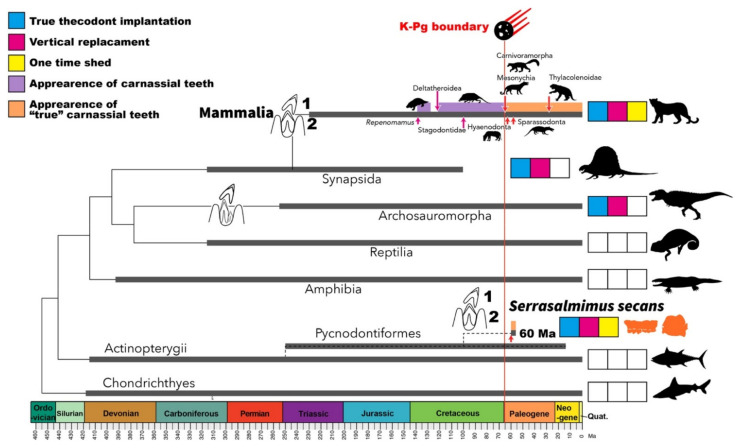
Dental features among vertebrates. A time-calibrated tree is a conceptual tree showing relationships of major vertebrate clades. The schematic teeth feature the hypothetical ancestral states of the replacement mode in major lineage. Blue: the presence of true thecodont implantation; magenta: vertical replacement system; yellow: two sets of teeth in a lifetime; orange: the appearance of carnassial teeth. All white boxes mean the absence of these characters. 1, 2: generation of a tooth. The oldest age of each taxon is followed by these data [Chondrichthyes, the earliest Devonian: [36]; Actinopterygii, the early Devonian: [37]; Amphibia, early Middle Devonian: [38]; Reptilia, the early Pennsylvanian, the late Carboniferous: [39]; Archosaur, the latest Permian: [40]; Synapsida, the early Pennsylvanian, the late Carboniferous: [41]; Mammalia, the late Carnian to the early/middle Norian, the late Triassic: [42]]. The images have the following credits: Eutheria by Synapsida by Dmitry Bogdanov (*Dimetrodon*: [43] under CC BY-SA 3.0 license), Archosauromorpha by Maija Karala (*Tyrannosaurus rex*: [44] under CC BY-SA 3.0 license), *Repenomamus* by Mateus Zica (*Repenomanus*: [45] under CC BY-SA 3.0 license), Mesonychia by Zimices (*Mesonyx*: [46] under CC BY-SA 3.0 license), Sparassodonta by Zimices (*Cladosictis*: [47] under CC BY-SA 3.0 license). The phylogenetic trees were illustrated using the geoscalePhylo function in the strap package [48] for the statistical programming language R [49].

## Data Availability

NMNS-PV20561 is kept at the National Museum of Natural History, Tsukuba, Ibaraki, Japan. All other data is listed in figures and tables.

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
