# Peer review of "A “Mammalian-like” Pycnodont Fish: Independent Acquisition of Thecodont Implantation, True Vertical Replacement, and Carnassial Dentitions in Carnivorous Mammals and a Peculiar Group of Pycnodont Fish"

_life, 2022, doi:10.3390/life12020250_

Round 1

Reviewer 1 Report

The manuscript is greatly improved after the revision. I detected only few issues in the Results/Discussion section which can be easily fixed. The authors state that the specialized carnivorous function has evolved by modifiying the plesiomorphic tribosphenic molars. However, as they state in the same paragraph, the adaptation to a carnivorous diet appears also in Eutriconodonta - however, these do not have tribosphenic molars, but pretribosphenic molars with three cusps in a row (triconodontan condition). This sentence should be reworded (see my suggestions in ms.).

In the next paragraph there is a sentence: "This period approximates the suggested duration of ecosystem restoration for macrofossils after the K.Pg event..." The word macrofossils should be replaced by e.g., metazoans, because the macrofossils did not live (fossils are dead) in the ecosystems.

See also minor suggestions for rewording in the same paragraph.

Otherwise the manuscript is now fine.

Author Response

Dear Reviewer 1

Thank you very much for your many valuable suggestions.

We accepted all the comments you mentioned in the pdf.

We feel the comments have helped us significantly improve the paper. Again, thank you very much for reviewingK our paper.

Best

Kumiko Matsui, Yuri Kimura

Reviewer 2 Report

Hi,

it is already better than the first version, but some minor details can still be improved.

I marked them in the pdf.

When those corrections are implemented, I do not need to see the ms again.

Best regards.

Author Response

Dear Reviewer 2

Thanks for your review.  We accepted almost all of your comments written in the pdf.

Title
We did not accept the suggestion of your title. Because mammalian-like is already used for non-fur animals, Synapsids (so-called mammalian-like reptiles). So, we were sure that this word doesn't mean furred animals, and followed the usage of it. 

Thank you again for your comments on our paper. Our manuscript was greatly improved for your kindful comments through the review process.

Sincerely,

Kumiko Matsui, Yuri Kimura

This manuscript is a resubmission of an earlier submission. The following is a list of the peer review reports and author responses from that submission.

Round 1

Reviewer 1 Report

Dear authors, I would like to congratulate you on the excellent finding and also on how you explored this finding. Macroevolutionary issues like the one studied in this manuscript are extremely interesting to science. I just missed a larger sample, but as the study was conducted based on fossils, which are extremely rare, I don't see this as an impediment to publication. Sincerely yours.        

Author Response

Dear Referee

Thanks for your polite review. We reviewed your comments in your commented pdf carefully.

  1. Italic letters
    As you mentioned, genera and species names in our MS were written in non-italic letters. These probably misconverted when the editorial office converted our manuscript to the format for review because when we re-checked our original manuscript, all species and genera names were written i italic letters. Thank you for your suggestion. We could check their mistakes before publishing.
  2. About the locality
    We don't have further specimen. So, we didn't add more detailed information and discussion about it.

Again, thank you for your kind review.

Sincerely yours, Kumiko and Yuri

Reviewer 2 Report

The observation on mode of tooth implantation and replacement in Serrasalmimus, complements the description of the taxon by Vullo et al (2017) and certainly is worth to be published. The presentation of the results is clear, the figures are of good quality.

The discussion and interpretation, however, are largely not supported by the data and need revision. The authors state that the evolution of carnassial teeth occurred contemporaneously in mammals (Carnivora) and in pycnodont fish, and they connect this to climatic changes (global warming) and the occupation of new ecological niches in marine and terrestrial environments. This interpretation is too far-fetched and is not supported by the fossil record. Carnassial teeth and carnivorous adaptation occurred already in Jurassic-Cretaceous eutriconodontan mammals, as did the vertical tooth replacement. The evolution of a carnassial-like dentition in an apparently highly specialized group of pycnodont fish cannot be used as evidence that actinopterygian fishes filled the now empty niches of marine top predators. This entire discussion should be removed from the manuscript and it should be focused on the specialization of tooth function and replacement within the pycnodont fishes. Please also change the title of manuscript accordingly. The statement on only one dental replacement cycle in Serrasalmimus should be rephrased in a more cautious manner because it cannot be excluded that there were more than one.

See Vullo et al. (2017), where a brief discussion on the various dental and dietary adaptations of pycnodonts is provided. This will make the manuscript much more solid and avoid stagy statements. Please see the attached annotated manuscript with specific comments.

Thomas Martin

Author Response

Dear Reviewer 2,

Thank you very much for providing many important insights. We are thankful for the time and energy you expended. Our responses to your suggestions are as follow:

General
We edited our manuscript following your kindful comments. After revising our manuscript, we received an English proofing from a native speaker following the instructions of the other referee. Therefore, some sentences might be changed by her.
As you mentioned, we wrote genera and species names in our MS in non-italic letters. These probably misconverted when the editorial office converted our manuscript to the format for review. When we re-checked our original manuscript, we wrote all species and genera names in italic letters. We could check their mistakes before publishing. Thank you for your suggestion. 

Discussion section
We largely modified the discussion section following your comments.
We deleted a discussion about global climate change. We included some discussions about Cretaceous mammals. In addition to that, we edited discussions about niche replacement and added some suggestions about sea "true" predators, Archaeoceti.

Again, thank you for giving us the opportunity to strengthen our manuscript with your valuable comments. 

Sincerely yours,
Kumiko Matsui
Yuri Kimura

Reviewer 3 Report

Dear authors and editors,

principally, this can be an interesting paper, but I have a couple of concerns:

  1. Why can't this be a mammal jaw? Please provide evidence that this is a pycnodontid. If you fail in doing so, the paper is obsolete.
  2. Provide a figure showing the differences in tooth replacement in the mentioned groups.
  3. What is the point of suppl. fig. S4? It is explained nowhere.
  4. The English is poor and needs revision.
  5. Referencing is poor. Many statements lack supporting evidence or references. See annotated ms.
  6. The origin of the phylogeny in Fig. 2 is unclear. In particular, the origins of actinopts and chondrichthyans are misplaced in the Silurian (see Brazeau & Friedman, 2015, The origin and early phylogenetic 
    history of jawed vertebrates, Nature, doi:10.1038/nature14438).
  7. Otherwise, the ms is concise and the figures are nice.
  8. Homoplastic and convergence are more or less synonyms. Choose one.

These are the main points.

Best regards.

Author Response

Dear Reviewer3

We sincerely thank you for your comments, which have helped us to improve our manuscript greatly.

General
We accepted almost all the suggestions and largely modified the Discussion section following your and other referee' suggestions.

English language and Style
Before we submitted, we had already received English proofing by Dr. Nicholas Pyenson, a curator of the National Museum of Natural History. We believed his proofing largely improved our English, but in addition to his proofing, we got the English revision service before we re-submitted our revised ms. Thank you very much for your helpful advice.

Italic letters
Thank you for your suggestion. As you mentioned, we did not write all genera and species names of our MS in italic letters. These probably misconverted when the editorial office converted our manuscript to the format for review. When we re-checked our original manuscript, all species and genera names were italicized. We could check their mistakes before publishing. 

Title
Thanks for your suggestion about our title. Following your suggestion, we changed our title. Thank you for your good suggestion.

Introduction
Thanks for your polite review in the Introduction session. We didn't add why Serrasalmimus secans could be identified as Pycnodontiformes. This manuscript does not describe new species or new specimens of Pycnodontiformes. We described the tooth replacement system of S. secans in it. Therefore, the reason why Vullo et al. (2017) identified S. secans as Pycnodontiformes is out of scope. Instead of this description in the Introduction section, we included a detailed description and diagnosis of S. secans in Supplemental Information.

Methods
Following your suggestion, we deleted a repeated sentence from MS.

Results and Discussion
We added many references you mentioned and deleted and modified some discussions.
              We largely revised the third paragraph following your mention "revise sentence structures". In addition, you mentioned "Far-fetched" in the last sentence of this paragraph, so we largely changed this sentence and added many references to strengthen our discussion.
              You suggested changing fresh-eating to carnivorous in the last paragraph. However, carnivorous are animals that eat various types of food related to animal bodies. We discuss hyper carnivorous animals with killing animals. I am so sorry, but we cannot accept your suggestion. You also mentioned that there is no evidence of "gene expression and regulation". However, we mentioned the difference in developments of dental tissues in mammals and fish in the 4th paragraph. To avoid misunderstandings, we have added some references here. We could promote our manuscript for your great suggestion. Thank you so much.

Figures
We carefully revised our figures based on your comments. We followed the earliest age of chondrichthyan from Miller et al. (2003, Nature). Our current legends are easy to lead to misleading, so I added references to the data source of the figure. Thank you very much for your polite review and suggestions.

Supplementary
We revised our supplementary based on your suggestion. However, we didn't delete the table of measurements (Table S1) because this data is not related to the description of teeth. However, measurements are essential data of fossils. Therefore, we decided to leave it. In Table S2, you suggested changing the table format; however, of course, you know, this manuscript is still under review, and I still don't know it will be able to accept for this journal. After getting the minor revision or acceptance, we will modify this table. I'll keep it in mind until then. Figure S4 is the result related to the supplementary file. We cited figure 4 in the section "Discussion about eruption of the replacement teeth in Serrasalmimus secans ".

Again, thanks for your polite review.

Sincerely,
Kumiko Matsui
Yuri Kimura